# Sleep as a Novel Biomarker and a Promising Therapeutic Target for Cerebral Small Vessel Disease: A Review Focusing on Alzheimer’s Disease and the Blood-Brain Barrier

**DOI:** 10.3390/ijms21176293

**Published:** 2020-08-31

**Authors:** Oxana Semyachkina-Glushkovskaya, Dmitry Postnov, Thomas Penzel, Jürgen Kurths

**Affiliations:** 1Department of Human and Animal Physiology, Saratov State University, Astrakhanskaya Str. 83, 410012 Saratov, Russia; postnovdmitry@googlemail.com (D.P.); thomas.penzel@charite.de (T.P.); juergen.kurths@pik-potsdam.de (J.K.); 2Physics Department, Humboldt University, Newtonstrasse 15, 12489 Berlin, Germany; 3Advanced Sleep Research GmbH, 12489 Berlin, Germany; 4Charité-Universitätsmedizin Berlin, Sleep Medicine Center, Charitéplatz 1, 10117 Berlin, Germany; 5Potsdam Institute for Climate Impact Research, Telegrafenberg A31, 14473 Potsdam, Germany

**Keywords:** Cerebral small vessel disease, sleep, slow wave activity, Alzheimer’S disease, blood-brain barrier

## Abstract

Cerebral small vessel disease (CSVD) is a leading cause of cognitive decline in elderly people and development of Alzheimer’s disease (AD). Blood–brain barrier (BBB) leakage is a key pathophysiological mechanism of amyloidal CSVD. Sleep plays a crucial role in keeping health of the central nervous system and in resistance to CSVD. The deficit of sleep contributes to accumulation of metabolites and toxins such as beta-amyloid in the brain and can lead to BBB disruption. Currently, sleep is considered as an important informative platform for diagnosis and therapy of AD. However, there are no effective methods for extracting of diagnostic information from sleep characteristics. In this review, we show strong evidence that slow wave activity (SWA) (0–0.5 Hz) during deep sleep reflects glymphatic pathology, the BBB leakage and memory deficit in AD. We also discuss that diagnostic and therapeutic targeting of SWA in AD might lead to be a novel era in effective therapy of AD. Moreover, we demonstrate that SWA can be pioneering non-invasive and bed–side technology for express diagnosis of the BBB permeability. Finally, we review the novel data about the methods of detection and enhancement of SWA that can be biomarker and a promising therapy of amyloidal CSVD and CSVD associated with the BBB disorders.

## 1. Sleep as a Potential Biomarker of Alzheimer’S Disease

Why do we need to sleep and how long should we sleep? Such very highly active people as Margaret Thatcher resent the idea of spending one third of their lives asleep and train themselves to get by with significantly less sleep than others. The surrealist painter Salvador Dali claimed to sleep for only three or four hours every night and to compensate for this by taking short naps during the day. On the other hand, Einstein liked to sleep about 14 h a day. However, regardless of the regime of sleep, without it, we become tired, our brain functions less well and prolonged sleep deprivation can be fatal. Indeed, the brain eats itself after short and chronic sleep loss via microglial activation and astrocytic phagocytosis of synaptic elements [1]. Insufficient sleep leads to sterile inflammation in the absence of infection [2,3,4] and to enhanced permeability of the blood–brain barrier (BBB) [3,5]. The total sleep deprivation of rats produced their death [6]. In humans the longest time of awakeness of 11 days is accompanied by hallucination and various cognitive deficiencies [7]. Thus, it seems obvious that sleep plays an important role in restoration of brain functions. However, what exactly is being restored by sleep remains unanswered. The functions of sleep have been speculated in the ancient works such as “Aristotle’s Theory of ‘Sleep and Dreams’” [8]. Aristotle proposed that sleep helps the body cleans its blood at the end of the day. More than 2000 years later, researchers confirmed Aristotle’s idea that sleep has a crucial function of clearance of metabolites and neurotoxic wastes from the brain accumulated in the awake central nervous system (CNS). So, Xie et al. demonstrated that the CSF tracer influx into the mouse brain is largely reduced by 95% in the awake state [9]. However, during deep sleep the brain’s interstitial fluid (ISF) volume expands (compared with wakefulness) by 60% via astrocytic aquaporin-4 (AQP4) water channels, resulting in faster waste removal, including toxins such as beta-amyloid (Aβ). The Fultz et al. in human studies discovered the close correlation between oscillation of CSF, which clears metabolic waste products from the brain, and EEG delta band during deep sleep [10].

Measuring how people sleep can be a promising approach to screen for Alzheimer’s disease (AD) [11,12,13]. People with AD have poor sleep, they often wake up and their nights become less refreshing as memory loss and other symptoms worsen [14,15,16]. The poor sleep quality and short sleep duration are associated with increased Aβ deposition Clinical studies have shown that Aβ content in CSF is lower in sleep than wakefulness There is evidence that Aβ clearance is increased during sleep due to increased ISF bulk flow [9]. Furthermore, excessive daytime sleepiness in older adults is associated with increased longitudinal Aβ accumulation [17]. Thus, there are growing body of evidence that disturbance of cleaning processes during sleep is a putative marker of AD pathology, at least in part, via an Aβ mechanism [18]. However, not all sleep can be an effective marker of AD.

Sleep consists of rapid eye movement (REM) associated with dreams and non-rapid eye movement (NREM) or deep sleep. REM sleep is characterized by desynchronization of EEG dynamics with faster oscillations and low voltage waveforms [19]. Human NREM sleep is subdivided into four stages and is defined as synchronous of EEG activity including sleep spindles (12–14 Hz) or K-complex (stage 2) and slow wave activity (0–4 Hz) in delta band (stage 3) [20]. The slow wave activity (SWA) is a major rhythm of deep sleep. The SWA is strongly controlled and a deficit of sleep induces a compensatory increasing of the SWA time [21,22]. Conversely, preceding daytime nap is accompanied by a reduction of the SWA time during subsequent nocturnal sleep [22]. The nature of SWA is not recognized yet, but there are growing body of evidence that SWA plays an important role in regulation of quality of sleep and in its restorative and clearing functions [10,23,24,25,26,27,28,29]. The sleep efficiency and depth are directly related to SWA [25,26,30,31,32,33]. NREM SWA considered as a promising intervention target for AD [23]. SWA sleep plays crucial role in memory consolidation and the SWA disturbances are associated with AD in patients and animal models [23]. Studies in animal models have found decreased SWA in P301S tau transgenic mice [34]. Both young and adult mice with a model of amyloidosis (APPswePS1dE9) demonstrate decrease in the cortical SWA power but not frequency with significantly reducing the time of NREM sleep [35,36]. Tg2576 and 3xTg-AD mouse models are characterized by the low time of SWA [37,38].

In humans, atrophy and Aβ accumulation in the medial prefrontal cortex is correlated with both decreased NREM-SWA and impaired overnight hippocampus-dependent memory consolidation in cognitively normal older adults [39,40]. Lucey et al. reported relationship between NREM-SWA and AD pathology, particularly tauopathy, and that this association was most evident at the lowest frequencies of NREM-SWA [11]. The changes in NREM-SWA, especially at 1 to 2Hz, might be able to discriminate tau pathology and cognitive impairment either before or at the earliest stages of symptomatic AD. The SWA disruption was also reported in patients with mild cognitive impairments [41]. The impairment of quality of sleep in cognitive normal old people could predict Aβ and tau accumulation in the brain. [42]. Taking into account all of the above, we suppose that slow wave sleep may be potential biomarker of AD that also have been discussed in other reviews [23,42].

Why SWA reflects AD pathology remains unknown but there is strong evidence that it can be related to recently discovered correlation between SAW and activation of glymphatic clearance during deep sleep [24]. Neither alpha nor gamma wave dynamics correlated with glymphatic funtions. The SWA contributes to the efficiency of fluid influx into the brain and clearance of waste and Aβ from the brain [9,43]. The reduced glymphatic perivascular flow with aging may facilitate the development of AD [44,45] due to the slower transit time that will cause greater cellular binding/update of Aβ and apolipoprotein E (apoE) [46]. The decreased glymphatic fluid transport after insufficient sleep may be related to an increase interstitial noradrenaline (NE) level [9,47,48,49] leading to NE-mediated decrease of astrocytic volume [50] and vasoconstriction of pial arteries [51]. Thus, the impaired CSF and ISF flow during sleep deficit can contribute to the reduced glymphatic fluid transport.

There is intrigue idea that the body posture during sleep can be used as an additional sign in diagnosis of impairment of glymphatic functions [52]. There is hypothesis that the most popular sleep posture (lateral) has evolved to optimize waste removal, including Aβ, during sleep and that posture must be considered in diagnostic imaging procedures developed in the future to assess CSF-ISF transport in humans [52].

Aβ plagues target synapses, contributing abnormalities in excitatory and inhibitory neurotransmission leading neural network disruption that can be responsible for reducing the time of SWA [35,53,54,55,56]. Indeed, more than 20% of cortical neurons exhibit hyperactivity surrounding Aβ [35] and blocking of neuron depolarization by the gamma-aminobutyric acid A (GABAA) improves SWA deficit in mice with AD [35,54]. The AAP mice demonstrate the deficit of the excitatory neurotransmitter glutamate and SWA, while the glutamate receptor antagonist alleviated hyperactivity and restored SWA [35,54]. Optogenetically increasing neural activity in hippocampal causes elevation of level of Aβ in ISF and Aβ depositions in the brain in mice with AD as well as augmentation of neural calcium content and decrease in the synaptic spine density [35,36,57]. Optogenetic-induced neural hyperactivity is accompanied by elevated release and propagation of tau in htau mice [58]. Taken together, Aβ-mediated synaptic inhibition, which leads to neural hyperactivity, can be another possible mechanism underling the SWA deficit in AD.

The aberrant astrocytic activity can also contribute to the SWA disruption in AD. Astrocytes maintain glutamate and GABA recycling and form the tripartite synapses to regulate synaptic transmission via calcium signaling [59,60]. Aβ depositions disrupt astrocytic morphology, astrocytic calcium signaling and glutamate/GABA recycling [61,62,63]. The blocking of astrocytic calcium transients resulted in decrease the number of astrocytes and neurons participating in regulation of SWA oscillations [64]. The astrocytic-mediated modulation of slow oscillations via intracalcium transient and extracellular glutamate triggers SWA [65,66]. Thus, the astrocytic abnormalities induced by Aβ accumulation in the brain may contribute to aberrant neural firing and lead to the hyperactivity of neurons, thus perturbation SWA. The further animal studies of the role of astrocytes in the SWA deficit in AD can shed light on mechanisms of SWA disturbances in AD and also might be point to a novel therapeutic methods for AD. Thus, sleep is natural factor, which activates clearance of accumulated metabolites from the brain, including Aβ, via increasing interstitial space and connective flow. Sleep disorders, especially SWA, are closely associated with reducing of glymphatic clearance of metabolites and toxins such as Aβ that can be an important informative platform for development of new promising strategies in early diagnosis of AD (Figure 1).

## 2. Slow Wave Activity as a Biomarker of Disruption of Blood-Brain Barrier

Sleep and the blood–brain barrier (BBB) are two important gamers in scenarios of the homeostasis of the central nervous system (CNS). Sleep is essential for maintenance of the health of the CNS via clearance of metabolites and neurotoxic wastes from the brain [9,10].

It is generally accepted that the BBB acts as the blood–brain interface protecting the CNS from the penetration of microorganisms and toxins from the blood. Therefore, the methods for opening the BBB are usually used for brain drug delivery and therapy of CNS diseases [67]. However, the latest findings changed our understanding of the role of BBB in the keeping of the CNS health. The BBB opening is accompanied by activation of clearance of macromolecules from the brain [68,69]. It can explain why the BBB opening without pharmacological therapy contributes for the clearance of Aβ in patients with AD and in mouse models of amyloidosis [70,71,72,73]. The interrelation between sleep and the BBB opening is not known but both conditions are interlinked with activation of clearance of macromolecules and toxins from the brain [9,68,69]. Thus, neurological activity during sleep is expected to be similar those during the BBB opening. Indeed, both sleep [9] and the BBB opening [70,71,72,73] are accompanied by clearance of Aβ from the brain. The sleep is characterized by the coupled oscillations of SWA and CSF, which cleans metabolic waste products from the brain [9,10], that becomes stronger with more low frequency EEG oscillations [24]. The SWA in EEG dynamics is also associated with BBB opening in humans and animals [74,75,76] and the BBB disruption causes activation of clearance of macromolecules from the brain [68,69].

The mechanisms of BBB-mediated changes of neural activity are not fully understood. The BBB opening can affect EEG activity by direct and indirect ways. Direct influence of an increased BBB permeability on the EED dynamics is generation signals of the BBB via electrophysiological properties of brain endothelial cells forming the BBB. The signals generated by the BBB originate from a trans-endothelial voltage between blood and brain tissue This voltage is a consequence of unequal endothelial cell apical and basolateral membrane potentials [77]. The ion influx/efflux changes the BBB permeability via brain endothelial cells membrane depolarization affecting the cell stiffness via molecular mechanisms underlying cortical actin cytoskeleton [77,78]. These changes of cell potential cause up to mV-level shifts in human scalp EEG [78,79]. Kiviniemi et al. observed that the intact BBB maintains a positive voltage, while the BBB leakage is characterized by a negative shift in this parameter [74].

In the 1970s, it was discovered that large-amplitude brain-potential shifts originate from a potential difference, which can occur during the BBB opening induced by respiratory acidosis in different animals species, including cats, money, and rats [80,81,82]. On the one hand, there is evidence suggesting that the BBB acts as a non-neuronal signal generator of mV-level slow shifts measured at scalp [79,82]. On the other hand, the BBB-signals can also be coupled to neuronal function, since low level frequency oscillations in the human brain are synchronized with faster cortical EEG oscillations and they are associated with the slow fluctuations in brain excitability [79,83,84].

Indirect influences of the BBB on the EEG behavior are astrocytes which are essential for the formation and maintenance of the BBB. Reduction in astrocyte number in the mPFC was associated with impaired cognitive flexibility and reduced power across delta (1–4 Hz), alpha (12–20 Hz), and gamma (30–80 Hz) frequency ranges [85,86]. The astrocytic mechanism of EEG modulation can be mediated via astrocyte-related regulation of the synaptic conductance [87,88,89], which are involved in electrically induced EEG-activated states in cortical neurons

We consider that the clearance of different compounds from the brain can be possible bridge between the similar changes in EEG dynamics co-occur during sleep and the BBB opening (Figure 2). It is believed that an increase in the volume of the interstitial fluid (ISF) contributes by drainage of water-soluble metabolites from ISF to CSF compartments [90].The sleep is associated with an increase in the ISF volume that is accompanied activation of macromolecular diffusion in the brain tissues [9,91]. The astrocytes rapidly and significantly change their volume, making a decisive contribution to the change in the total proportion of volume of ISF [92,93,94]. These dynamic astrocyte volume changes may represent a previously unappreciated yet fundamental mechanism by which astrocytes regulate brain rhythm during sleep [95,96] and the BBB integrity [97,98].

The SWA would be the cleaning power of sleep [24]. Indeed, during awake stage and REM sleep, the brain’s rhythms are desynchronized due to cacophony of neuron activity connecting different parts of brain [19,20]. During SWA sleep, the neuron dynamics demonstrates synchronous activity and inactivity changing each other over brief periods. We think that these oscillations of synchronous activity and inactivity of delta rhythm may be helping in “brain rinsing” and to move brain fluids and waste products through brain tissues like sea waves move salt and water. The reducing of slow wave sleep might be one of important diagnostic symptom of altered clearance of the brain that can contribute development of neurodegenerative diseases via accumulation of toxins in the brain [11,12].

## 3. Slow Sleep Wave Enhancement Is Promising Therapy of Alzheimer’S Disease

Current, there are no therapy of AD [99,100,101]. The majority of clinical approaches are focus on using monoclonal antibodies as passive immunotherapy [102]. However, pharmaceutical companies such as Biogen, Johnson & Johnson, Pfizer announced the cancellation of funding for the synthesis of antibodies for the treatment of AD due to the failure of clinical trials (Biogen/Eisai Halt Phase 3 Aducanumab Trials. https://www.alzforum.org/news/research-news/biogeneisai-haltphase-3-aducanumab-trial).

Obviously, in the next couple of decades, the main strategies for a treatment of AD will be non-invasive methods of stimulation of clearance of the toxic Aβ from the brain. The enhancement of SWA sleep is discussed as a promising tool in therapy of AD and rescue sleep-dependent memory consolidation [23].

Auditory stimulation of SWA appeared to be more effective in increasing SWA and improvement of memory consolidation as well as cognitive functions [103,104,105,106,107]. This method is based on uses of “pink noise” (50-millisecond bursts) that is synchronized with neural cortical activity of delta band and increases the time of SWA [105,106,107]. The morphology, topography and propagation pattern of auditory-stimulated SWA are very similar to those of SWA observed during natural sleep [105,106].

Auditory stimulation is used in overnight and nap studies [107,108,109,110,111]. Overnight studies began 5 min after falling into NREM for the first time and ended 210 min later [109,110,111]. In afternoon nap studies, auditory stimulation is used intermittently regime with 90-min nap session [107,112,113]. The auditory-mediated SWA enhancement is hypothesized to be the results of “bottom up” activation of large populations of cortical neurons as the same process that is underlying arousing the organism The intensity of sensory stimulation has to be strong enough to trigger SWA enhancement, but no so strong to cause awakening. Thus, the intensity of stimulation has to be strong enough to trigger the activation of the reticular ascending system (ARAS) playing a crucial role in arousal, but not so strong as to cause a full-blown awakening. The idea that the arousal systems can be functionally parceled according to the magnitude of stimulation was first by Moruzzi in 1950s. He considered that for mild sensory stimulation only some portions of the activating ARAS might be activated, while the entire system could be recruited only by more intense stimuli [114,115].

Therefore, the optimization of acoustic stimulation of SWA such as intensity, frequency, and timing is in the trend of development of breakthrough technologies in the enhancement of SWA [103,104,105,106,107].

Transcranial electrical (tDCS) and magnetic stimulation have been successfully applied to enhance SWA with the aim to improve quality of sleep and behavior/cognitive functions [116,117,118,119,120,121,122]. tDCS (1–20 Hz) triggers SWA that is indistinguishable from those during natural sleep [118,121]. tDCS induces a widespread electrical potential field with a focus on fronto-cortical areas. In majority of studies, tDCS is delivered at 0.75 Hz for 5 min intervals separated by 1 min off periods after SWA onset [116,117]. However, the long-term effects of repeated exposure of both these methods remain unknown [105]. The complex of response of activated/deactivated cortical fields following two these methods and difficulties of characterization of precise mechanism of tDCS and magnetic stimulation make unpredictable effects of these approaches. Since these methods are currently impractical and their safety is questionable, especially for chronic long-term exposure, natural physiological sensory stimulation of SWA is more preferable.

Pharmacological methods can also lead increase the time of SWA and might be alternative strategies relying on electric, magnetic, or auditory stimulation of SWA for improving of quality of sleep and brain functions. The administration of tiagabine [123,124], gaboxadol [125], sodium oxybate [126,127,128], baclofen [128], olanzapine [129,130], interleukin-6 [131] has been demonstrated as pharmacological method for SWA enhancing. Tiagabine, gaboxadol, sodium oxybate and baclofen increases SWA via the inhibition of neurotransmitter GABA [132]. Olanzapine is an antagonist of the serotonin2C (5-HT2C) receptor, which is involved in SWA regulation [133]. Interleukin-6 as proinflammatory cytokinine stimulates neuromedulatory mechanisms of regulation of SWA nature [134]. The pharmacological SWA enhancement is not new idea and were discussed in these reviews [134,135,136,137,138]. Anesthesia such as ketamine/xylazine or dexmedetomidine increase in EEG delta power [139,140] and significantly enhance glymphatic influx [24].

Optogenetic modulation of SWA is a leading-edge research method that can be used for restoring brain oscillatory brain activity, including SWA. This method is based on light cell-targeting manipulation of proteins expression in cells leading to modulation of the neural activity within neural circuits interest. Optogenetic-mediated activation of neural nitric oxide synthase (nNOS) or somatostatine neurons is useful for restoring SWA [141]. The optogenetically evoked responses in nNOS-positive cells of the cerebral cortex improve memory, sleep quality and prolong the time of SWA [142]. Thus, SWA restoration provides a promising novel therapeutic target for AD. Development of breakthrough strategies targeting SWA during NREM sleep might be a promising therapeutic tool to slow memory decline in the elderly or in healthy individuals at risk for developing AD as well as to delay progression of the disease in patients with AD (Figure 3).

## 4. Conclusions

Cerebral small vessel disease (CSVD) is an important course of cognitive decline and AD. An key pathophysiological mechanism of CSVD is BBB leakage leading to progression of CSVD. Currently, there are no specific preventive or therapeutic measures to improve CSVD. Sleep can be a novel biomarker and a promising therapeutic target for amyloid CSVD and for CSVD associated with BBB disruption. Amyloid CSVD such as AD is correlated with SWA deficit at early stages of disease and was found in asymptomatic cognitively normal adults. Since SWA disruption is an early event, it can be an early biomarker for AD. The level of Aβ in the brain depends on quality of SWA and progression of AD is associated with reducing the time of SWA. Thus, the SWA pattern has the potential to be used as prognostic tool of severity of AD. The latest findings clearly suggest that SWA can be pioneering non-invasive and bed–side technology for express diagnosis of BBB leakage. Thus, SWA restoration provides a promising novel therapeutic target for amyloid CSVD and for CSVD associated with the BBB disruption. A better understanding of the link between SWA disruptions and reducing of clearance Aβ from the brain associated with memory decline in AD may shed light on the mechanisms of AD. The methods of detection and enhancement of SWA during NREM sleep can open a new era in novel strategies of early therapy of CSVD, including improving memory in patients with AD or in healthy individuals at risk for AD.

## Figures and Tables

**Figure 1 ijms-21-06293-f001:**
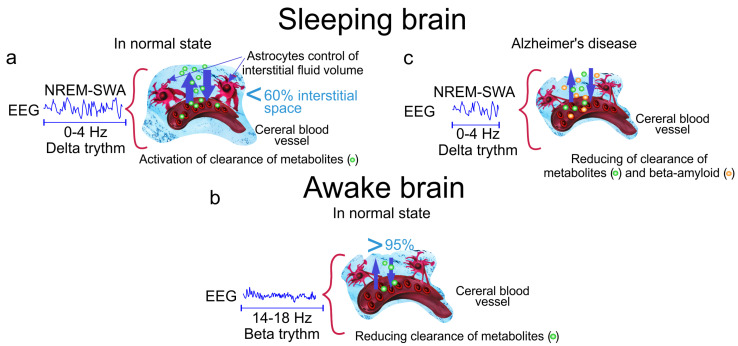
The cleaning power of a slow wave activity (SWA) during deep sleep. (**a**) The slow wave sleep is accompanied by increasing the interstitial fluid (ISF) volume by 60% via astrocytic-aquaporin (AQP)-channels that contributes augmentation of metabolic clearance; (**b**) Wakefulness reduces diffusion of metabolites by 95% via decreasing the ISF volume; (**c**) Alzheimer’s disease is associated with accumulation of Aβ in the brain tissues due to reducing of the time of SWA and suppression of clearance of toxic protein from the brain.

**Figure 2 ijms-21-06293-f002:**
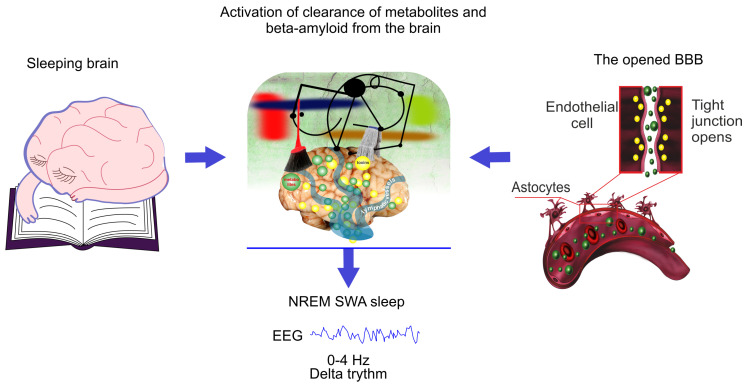
Schematic illustration of hypothesis that the EEG characteristics of non-rapid eye movement (NREM) SWA sleep is similar for natural sleep and the blood–brain barrier (BBB) disruption due to the same mechanism of activation of clearance of metabolites and toxins such as Aβ from sleeping brain and from the areas surrounding the opened BBB.

**Figure 3 ijms-21-06293-f003:**
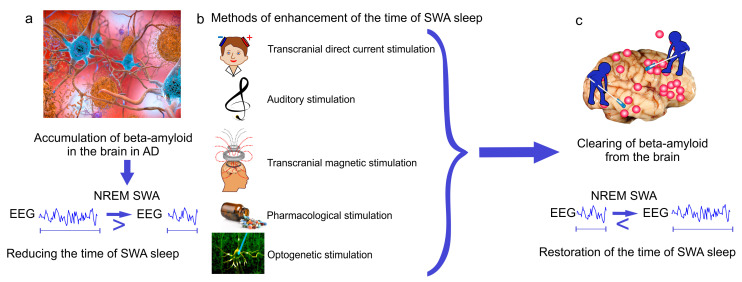
New breakthrough strategies targeting SWA sleep for therapy of Alzheimer’s disease (AD). (**a**) Alzheimer’s disease is characterized by Aβ-mediated reducing the time of SWA sleep; (**b**,**c**) methods of enhancement of SWA oscillations restore the time of SWA sleep and memory via improvement of glymphatic clearance of Aβ from the brain.

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
