# Peer review of "Sleep as a Novel Biomarker and a Promising Therapeutic Target for Cerebral Small Vessel Disease: A Review Focusing on Alzheimer’s Disease and the Blood-Brain Barrier"

_ijms, 2020, doi:10.3390/ijms21176293_

Round 1

Reviewer 1 Report

This reviewer has no further comments.

Author Response

We thank the referee for the careful and insightful review of our manuscript.

Authors

Reviewer 2 Report

Thank you for the opportunity to review your manuscript, entitled Breakthrough strategies of clearance of macromolecules and toxins from the sleeping brain via the lymphatics. The removal of macromolecules -- such as Aβ -- as well as neurotoxins associated with traumatic brain injury is essential to slowing the pathogenesis of neurogenerative conditions as well as reducing disease burden. To that end, recent work on the role of sleep in glymphatic clearing of toxins (e.g. Fultz et al, as referenced in the present work) represents a potentially critical understanding of this process and how some individuals may be resilient against such effects. In this respect, a review of the role of sleep in (g)lymphatic clearing as well as novel technology options for enhancing these effects is welcome.

However, this review falls short -- for me -- in this respect. Both the title and abstract suggest that the focus is on the sleeping brain and that the emphasis is on novel, groundbreaking methods to "overcome limitations limitations in the development of breakthrough technologies of night stimulation of lymphatic clearance of macromolecules and toxins" (your words, from the abstract).

In reality, this is a review of how the lymphatic system in the brain may work and the potential interplay with CSF production. There is a limited section (lines 140-167 in the produced PDF) addressing the role of sleep as well as a section on ultra-slow rhythms (lines 346-360). There is a statement in the conclusion, which I believe is the driving motivator and indeed truly intended emphasis of this review -- on lines 432-433:

"Sleep as the natural driving factor for activation of the cerebral LVs is a platform for development of innovative strategies in stimulation of drainage and cleaning function of LVs."

However, this idea is not developed in a discernible way in the manuscript. As I read this manuscript this is what I perceived/understood as relates to the stated objective of the manuscript:

Section 1 sets up the role of lymphatic clearance and sets the stage for how sleep may be a driving factor and I really appreciated the approach that section 1 takes. Section 2.2 then addresses the potential role of sleep itself (as we know it) and coupled with section 4.4 address how sleep may contribute to the pulsatory motion of CSF into/out of the lymphatic flow. Section 2.1 then follows onto this notion (though it is sequentially near the beginning) with a technological approach that may help to enhance CSF flow in sleep using phototherapy via LV relaxation and increased permeability. 

Sections 3, 4.1-4.3, 4.5, and 4.6, by contrast, have little to no connection to sleep nor how to overcome challenges in developing breakthrough technologies. For example (my words, my interpretation), how do the methods as well as challenges in measuring lymphatic flow rate in mouse ears move us toward broadly utilitarian methods of photostimulation during sleep in humans. The connection is not there. These sections are an excellent overview of the methods and theories of CSF flow quantification as well as the challenges but the connection to sleep and developing breakthrough technologies (the overarching intended focus of the review) is simply absent.

Additionally absent, and something I would have expected to see, is how to connect this to sleep-related interventions in humans. Considerable time is spent discussing animal models -- which is necessary and critical given that much of the ground-laying work is there. However, it is not evident how to take this knowledge and create technological breakthroughs that improve the quality of life/aging/injury recovery in humans.

I am very invested in the idea that sleep can be enhanced through technology to improve lymphatic clearing as a way to reduce long-term neurodegenerative burden and I see a review of that kind as necessary in providing a backbone for future work. However, in its present form, I find this review to lack a clear focus, connection to an underlying theme, and an alignment with the stated goals and conclusions. Please consider what the true purpose of this manuscript is and revise it to align that. If that means separating the review into two (i.e., one review on CSF movement and how lymphatic drainage in the brain may function and one on the role of sleep and potential technological avenues for enhancing sleep-related lymphatic clearing) so that important information and themes can be appropriately developed, then please consider doing so. 

On a technical note: Please clarify if the text description of Figure 5A and 5B (line 235-236), the figure, and the figure legend all agree as intended. Conflicting descriptions of Figure 5B are presented in line 235 sentence 1 and 235-236 sentence 1. The legend indicates a green and red line on 5A which are only present on 5B.

I look forward to reading a revision of this manuscript.

Author Response

We thank Reviewer for helprful comments and advices.

We received strong request from academic editor to change completely the idea and strategy of our review. He recommended us to write new manusript and devide it on two parts focising on 1) the newest aspect of glymphatics in the brain and in small vessel disease and 2) on sleep, glymphatic system, and SVD, i.e. to be focused precisely on the topic the special issue 2.0.

Therefore, we should remove many sessions of our review and to write new manuscript following request of acamedic editor.

The title of our new review is "Sleep as a Novel Biomarker and a Promising Therapeutic Target for Cerebral Small Vessel Disease:
A Review Focusing on Alzheimer’s Disease and the Blood-Brain Barrier".

We took into account your helpful advices in the preparation of the new version of manuscrip. In this review, we show the strong evidences that slow wave activity (SWA) (0-0.5 Hz) during deep sleep reflects glymphatic pathology, the BBB leakage and memory deficit in Alzeimer's disease (AD). We also discuss that diagnostic and therapeutic targeting of SWA in AD might lead to be a novel era in effective therapy of AD. Moreover, we demonstrate that SWA can be pioneering non-invasive and bed–side technology for express diagnosis of the BBB permeability. Finally, we review the novel data about the methods of detection and enhancement of SWA that can be biomarker and a promising therapy of amyloidal CSVD and CSVD associated with the BBB disorders.

We would like to thank Reviewer again for thoughtful comments and efforts towards improving our manuscript.

Authors

Round 2

Reviewer 2 Report

This is much more clear and focused. My only concern at this point is the readability. Thorough English editing and revision is necessary throughout.

This manuscript is a resubmission of an earlier submission. The following is a list of the peer review reports and author responses from that submission.

Round 1

Reviewer 1 Report

The authors have done the minimum to address our previous comments, despite our statement that there are numerous inconsistencies throughout the manuscript.

References are purposely misinterpreted to support the authors’ hypothesis that there are lymphatic vessels within the brain itself., e.g. regarding ref. 32 “found the lymphatic vessels in deep areas of the telencephalic hemispheres of healthy mice”. This study 1) used mice in their infancy (14 days old), 2) describes lymphatic vessels “in the dural septae entering into deeper parts of the brain”, i.e. in the meninges not the brain, as well as “Lyve-1-positive structures in deeper aspects of the telencephalic hemispheres.” Lyve-1 is notonly expressed by lymphatic vessels, but also by developing blood vessels and a subset of macrophages. If the authors wish their viewpoint to be seriously considered, they should present solid evidence in an unbiased fashion.

Statements such as “the anatomy of lymphatic pathway of clearance of macromolecules from the brain remains poorly understood” (line 47) suggest without evidence that such a pathway indisputably exists. The abstract states “we motivate readers for a substantial revision of our knowledge about the anatomy and physiology of the lymphatics of the CNS”. The authors should strive to present all sides of this controversial topic so that the reader can come to their own conclusions. 

There are too many instances to list where the text is unclear or misleading (either due to English language issues or the biased nature of the text). 

Author Response

We would like to express our gratitude to Reviewer for the critical comments and constructive advices. It is correct, Melanie Lohrberg and Jörg Wilting presented their results on 14 days old mice. However, Antila et al. (https://doi.org/10.1084/jem.20170391) demonstrate that the very first MLVs are observed just before birth at the foramen magnum (FM), and starting from postnatal day (P) 0, they extend, following major arteries and venous sinuses until the complete formation of the MLVs at P24. The authors also describe the development of the MLVs in the spinal cord, where it is initiated at P4 from the FM and is completed before P36. These facts allow to think that P14 can be enough to distinguish the lymphatic vessels. 

We are agree completely with Reviewer that there are only few hints demonstrating possibility of presence of the LVs in the brain. But, this hypothesis is discussed in scientific community. Koh et al. (2005) wrote in their review (https://www.ncbi.nlm.nih.gov/pmc/articles/PMC1266390) “Apart from one study in which Prineas reported what appeared to be lymphatic vessels within the brain parenchyma of individuals with neurological disorders, it is now accepted that lymphatic vessels do not exist in the brain and spinal cord”. Yes, Prineas described the lymphatic capillaries and lymphoid tissues in the human brain and spinal cord with neurological disorders without using the markers of lymphatic endothelium. However, Louveau et al.  also demonstrated electronic miscroscopy of the MLVs without special markers (https://www.nature.com/articles/nature14432). We discussed it personally with Jonathan Kipnis and he explained the specific ultrastructure, which typical only for the LVs. This means that Prineas presented the pioneering results of presence of LVs in the human brain with neurological pathology. Lohrberg et al. obtained other initial hints for the presence of lymphatics in the meningeal septae penetrating into the brain that can be bridge in our understanding how MLVs can communicate with the putative LVs in the brain parenchyma described by Prineas. We would to make these observations known to broader audience in order to motivate the scientists for the detailed study of the lymphatic structures in the CNS. If it will be confirmed, this could offer new approaches for studying the etiology and therapy of brain diseases. 

We have corrected and toned down the claims that the LVs might be presented in the brain in the text of manuscript. New changes are highlighted by red color. Please see the attachment. 

We thanks again Reviewer for the constructive advices and recommendations.

Authors

Reviewer 2 Report

In the review by Semyachkina-Glushkovskaya et al, the author are providing an opinion article on tools to measure and analyze lymphatic drainage from the CNS.

The authors have incorporated most of my comments from the first round of revision. Some elements are still missing in this reviewer's opinion. When talking about the presence of meningeal lymphatic vessels in human, the study by the lab of Dani Reich (Absintha et al) should be cited. 

In figure 1, the authors are using Lyve1 to identify the meningeal lymphatic vessels. Figure 1a is not convincing for lymphatic vessels. Indeed, Lyve1 is also expressed by multiple macrophages population in the CNS and the lack of lumen in Figure 1a is more indicative of a macrophage than a lymphatic vessel. If the authors want to claim for the presence of RBCs in the lumen of the meningeal lymphatic vessels after ICH, some quantification of number of luminal RBC would be necessary (especially to remove dissection artifact). This reviewer's understand that such experiment using human tissue is challenging. In this case, the statement made by the authors regarding RBCs drainage through the lymphatics should be toned down.

Author Response

We thanks Reviewer for the valuable suggestions and helpful recommendations. We cited the article of Seung-Kwon Ha et al.

It is correct that Lyve-1 can be expressed also in macrophages and in the cerebral vessels. We discussed these facts in our previous review (Int. J. Mol. Sci. 2018, 19, 3818; doi:10.3390/ijms19123818).  

Notice, there are some findings where expression of Lyve-1 was also observed in the human iliac atherosclerotic arteries [54], in the embryonic blood vessels [55], in macrophages [56], the reticulo-endothelial system [48]. Therefore, the specificity of Lyve-1 as a marker for the lymphatic vessels is not strong enough”. 

48. Zheng, M.; Kimura, S.; Nio-Kobayashi, J.; Iwanaga, T. The selective distribution of LYVE-1-expressing endothelial cells and reticular cells in the reticulo-endothelial system. Biomed. Res 2016, 37, 187–198. 

54. Grzegorek, I.; Drozdz, K.; Chmielewska, M.; Gomulkiewicz, A.; Jablonska, K.; Piotrowska, A.; Karczewski, M.; Janczak, D.; Podhorska-Okolow, M.; Dziegiel, P.; et al. Arterial wall lymphangiogenesis is increased in the human iliac atherosclerotic arteries: Involvement of CCR7 receptor. Lymphat. Res. Biol. 2014, 12, 222–231.  

55. Gordon, E.J.; Gale, N.W.; Harvey, N.L. Expression of the hyaluronan receptor LYVE-1 is not restricted to the lymphatic vasculature; LYVE-1 is also expressed on embryonic blood vessels. Dev. Dyn. 2008, 237, 1901–1909. 

56. Lim, H.Y.; Lim, S.Y.; Tan, S.K.; Jackson, D.C.; Ginhoux, F.; Angeli, F. Hyaluronan Receptor LYVE-1-Expressing Macrophages Maintain Arterial Tone through Hyaluronan-Mediated Regulation of Smooth Muscle Cell Collagen. Immunity 2018, 49, 326–341.  

In figure 1 we present the results of identifiсation of MLVs in the human dura matter using monoclonal Anti-LYVE1 antibody [EPR21857] ab219556. In these ICH analysis we observed the Lyve-1 positive vessels filled RBCs and the immune cells, including monocytes and macrophages, which were not labeled by Lyve-1.  

The lymphatic vessels rarely show the distinct lumen in the normal state, especially in the dura matter due to their very small size.   

We changed the photos in figure 1 a and b as well as we added quantification of the number of RBCs per mm of vessels in the Lyve-1 positive vessels and in the blood vessels with similar diameter. New changes are highlighted by red color. Please see the attachment. 

We would like to thanks again Reviewer for constructive comments and advices. 

Authors
